# Breast cancer genome and transcriptome integration implicates specific mutational signatures with immune cell infiltration

Marcel Smid[1,*], F. Germán Rodríguez-González[1,*], Anieta M. Sieuwerts[1], Roberto Salgado[2,3], Wendy J.C. Prager-Van der Smissen[1], Michelle van der Vlugt-Daane[1], Anne van Galen[1], Serena Nik-Zainal[4,5], Johan Staaf[6], Arie B. Brinkman[7], Marc J. van de Vijver[8], Andrea L. Richardson[9,10], Aquila Fatima[11], Kim Berentsen[7], Adam Butler[4], Sancha Martin[4], Helen R. Davies[4], Reno Debets[1], Marion E. Meijer-Van Gelder[1], Carolien H.M. van Deurzen[11], Gaëtan MacGrogan[12], Gert G.G.M. Van den Eynden[3,13], Colin Purdie[14], Alastair M. Thompson[14], Carlos Caldas[15], Paul N. Span[16,17], Peter T. Simpson[18], Sunil R. Lakhani[18,19], Steven Van Laere[20], Christine Desmedt[2], Markus Ringnér[6], Stefania Tommasi[21], Jorunn Eyford[22], Annegien Broeks[23], Anne Vincent-Salomon[24], P. Andrew Futreal[25], Stian Knappskog[26,27], Tari King[28], Gilles Thomas[29], Alain Viari[29,30], Anita Langerød[31,32], Anne-Lise Børresen-Dale[31,32], Ewan Birney[33], Hendrik G. Stunnenberg[7], Mike Stratton[4], John A. Foekens[1] & John W.M. Martens[1]

A recent comprehensive whole genome analysis of a large breast cancer cohort was used to link known and novel drivers and substitution signatures to the transcriptome of 266 cases. Here, we validate that subtype-specific aberrations show concordant expression changes for, for example, *TP53, PIK3CA, PTEN, CCND1* and *CDH1*. We find that *CCND3* expression levels do not correlate with amplification, while increased *GATA3* expression in mutant *GATA3* cancers suggests *GATA3* is an oncogene. In luminal cases the total number of substitutions, irrespective of type, associates with cell cycle gene expression and adverse outcome, whereas the number of mutations of signatures 3 and 13 associates with immune-response specific gene expression, increased numbers of tumour-infiltrating lymphocytes and better outcome. Thus, while earlier reports imply that the sheer number of somatic aberrations could trigger an immune-response, our data suggests that substitutions of a particular type are more effective in doing so than others.

[1] Department of Medical Oncology, Erasmus MC Cancer Institute and Cancer Genomics Netherlands, Erasmus University Medical Center, 3015CN Rotterdam, The Netherlands. [2] Breast Cancer Translational Research Laboratory, Université Libre de Bruxelles, Institut Jules Bordet, Bd de Waterloo 121, B-1000 Brussels, Belgium. [3] Department of Pathology/ TCRU GZA, 2610 Antwerp, Belgium. [4] Wellcome Trust Sanger Institute, Hinxton CB10 1SA, Cambridge, UK. [5] East Anglian Medical Genetics Service, Cambridge University Hospitals NHS Foundation Trust, Cambridge CB2 9NB, UK. [6] Division of Oncology and Pathology, Department of Clinical Sciences Lund, Lund University, SE-223 81 Lund, Sweden. [7] Faculty of Science, Department of Molecular Biology, Radboud Institute for Molecular Life Sciences, Radboud University Nijmegen, 6525GA, Nijmegen, The Netherlands. [8] Department of Pathology, Academic Medical Center, Meibergdreef 9, 1105 AZ Amsterdam, The Netherlands. [9] Department of Pathology, Brigham and Women's Hospital, Boston, Massachusetts 02115, USA. [10] Dana-Farber Cancer Institute, Boston, Massachusetts 02215, USA. [11] Department of Pathology, Erasmus MC Cancer Institute, Erasmus University Medical Center, 3015CN Rotterdam, The Netherlands. [12] Département de Biopathologie,Institut Bergonié, CS 61283 33076 Bordeaux, France. [13] Molecular Immunology Unit, Jules Bordet Institute, B-1000 Brussels, Belgium. [14] Department of Pathology, Ninewells Hospital & Medical School, Dundee DD1 9SY, UK. [15] Cancer Research UK Cambridge Institute, University of Cambridge, Li Ka Shing Centre, Robinson Way, Cambridge CB2 0RE, UK. [16] Department of Radiation Oncology, Radboud University Medical Center, 6525GA, Nijmegen, The Netherlands. [17] Department of Laboratory Medicine, Radboud University Medical Center, 6525GA, Nijmegen, The Netherlands. [18] The University of Queensland: UQ Centre for Clinical Research and School of Medicine, Brisbane 4029, Australia. [19] Pathology Queensland, The Royal Brisbane and Women's Hospital, Brisbane 4029, Australia. [20] Center for Oncological Research, University of Antwerp & GZA Hospitals Sint-Augustinus, 2610 Wilrijk, Belgium. [21] IRCCS Istituto Tumori 'Giovanni Paolo II', 70124 Bari, Italy. [22] Cancer Research Laboratory, Faculty of Medicine, University of Iceland, 101 Reykjavik, Iceland. [23] The Netherlands Cancer Institute, 1066CX Amsterdam, The Netherlands. [24] Department of Pathology and INSERM U934, Institut Curie, 26 rue d'Ulm, 75248 Paris Cedex 05, France. [25] Department of Genomic Medicine, UT MD Anderson Cancer Center, Houston, TX, 77230, USA. [26] Department of Clinical Science, University of Bergen, 5020 Bergen, Norway. [27] Department of Oncology, Haukeland University Hospital, 5021 Bergen, Norway. [28] Memorial Sloan Kettering Cancer Center, 1275 York Ave, New York, New York 10065, USA. [29] Synergie Lyon Cancer,Centre Léon Bérard, 28 rue Laënnec, Cedex 08 Lyon, France. [30] Equipe Erable, INRIA Grenoble-Rhône-Alpes, 655, Av. de l'Europe, 38330 Montbonnot-Saint Martin, France. [31] Department of Cancer Genetics, Institute for Cancer Research, Oslo University Hospital The Norwegian Radiumhospital, 0310, Oslo, Norway. [32] K.G. Jebsen Centre for Breast Cancer Research, Institute for Clinical Medicine, University of Oslo, 0310 Oslo, Norway. [33] European Molecular Biology Laboratory, European Bioinformatics Institute, Wellcome Trust Genome Campus,Hinxton CB10 1SD, Cambridgeshire, UK. * These authors contributed equally to this work. Correspondence and requests for materials should be addressed to J.M. (email: j.martens@erasmusmc.nl).

The recent advance in DNA sequencing technologies has added substantially to the knowledge-base of breast cancer, but also reaffirmed its proverbial heterogeneity[1–8]. Sequencing DNA to greater depth enables analyses in distinguishing the 'driver' mutations—thought to be involved in the tumourigenesis—from the 'passenger' mutations—those arising in the lifetime of the cancer cell, without affecting the cancer cell's fitness. We recently reported on the identification of 12 substitution and 6 rearrangement signatures, as well as 7 consensus patterns therein, using whole genome sequence (WGS) data of 560 cases with primary breast cancer[9]. Numbered according to an earlier scheme[2], some of the 12 substitution signatures had underlying mechanisms likely explaining the signature: for example, signature 1 was described as 'age-related' due to the many C>T mutations in an NCG trinucleotide context (the underlined base is mutated). Similarly, signatures 2 and 13 were related to APOBEC-type mutations (predominant C>T and C>G in a TCN context, respectively), while signature 3 lacked specific features but was strongly associated with inactivated BRCA1 and BRCA2. Signature 5 (with unknown causal mechanism) is present in most cases and is primarily characterized by C>T and T>C mutations. Other signatures were found in few cases (for example, signatures 6, 20 and 26) showing signs of mismatch repair deficiency[9]. It is important to realize that most of the breast cancer cases exhibit a blend of these signatures, possibly obfuscating the resulting effects. The very diverse pattern of mutations found in primary breast cancer was previously studied in depth[1] in a large cohort, integrating (epi)genomic, transcriptomic and proteomic data, but this study mainly focused on the driver genes and reported the resulting effect on its expression in solo or in concert in pathway analyses. The possible effects of the mutational signatures themselves on the transcriptome have been studied with less scrutiny.

In the current study we analyse the transcriptome of 266 cases (191 ER-positive/Her-2 negative and 75 triple-negative) by RNA sequencing with available WGS results[9] to explore the possible consequences of the substitution signatures at the transcriptome level. We report on clinically and biologically relevant gene expression signatures and associate these with the number and character of signature mutations.

## Results

**Validation of cohort.** RNA-sequencing data of 266 primary breast cancer cases with available WGS data were generated and were first analysed and visualized to verify whether the cohort exhibited the archetypical breast cancer patterns. To this end, the 5,000 most variable transcripts were used to correlate and hierarchically cluster all cases resulting in five clusters (Fig. 1). Cluster 1 contained all cases of the basal-like intrinsic subtype[6,10] and held 92% of ER-negative cases, while the remaining four subclusters (clusters 2–5) were luminal. Among the subtypes, the total number of substitutions was highest in the basal-likes (Kruskal–Wallis (KW) $P < 0.0001$), while within the luminal cases, the more aggressive luminal B (refs 11,12) compared with luminal A type cancer also showed higher numbers of substitutions (Mann–Whitney $U$-test (MWU) $P < 0.0001$, Supplementary Fig. 1, and previously reported[1]).

Next, we evaluated the reported driver genes (derived from WGS data[9]) with at least five events in this cohort (with the top recurrent events presented in Fig. 1). Amplification of MYC ($n = 47$, $\chi^2$ $P < 0.0001$), of CCNE1 and CCND3 (both $n = 7$, $\chi^2$ $P = 0.0003$), of PIK3CA ($n = 9$, $\chi^2$ $P < 0.0001$) and, as expected, mutations in TP53 (refs 12–14) ($n = 105$, $\chi^2$ $P < 0.0001$) were predominant in basal subtype cancers, while mutations in GATA3 (refs 8,15) ($n = 25$, $\chi^2$ $P = 0.0006$), and PIK3CA ($n = 74$,

$\chi^2$ $P < 0.0001$) and amplification of CCND1 ($n = 39$, $\chi^2$ $P < 0.0001$) were largely restricted to luminal breast cancer, with the latter strongly related to luminal B cancers[16,17] (33/39 cases, $\chi^2$ $P < 0.0001$). The lobular cancer driver[18,19] CDH1 was mutated in lobular cases (15/21, $\chi^2$ $P < 0.0001$) and mutations of this type were predominantly observed in luminal A cases (12/21, $\chi^2$ $P = 0.0013$). In line with their oncogenic role, amplified regions had significantly higher expression for MYC, CCND1, ZNF217, PIK3CA, MDM2, CCNE1, IGF1R, KRAS and ERBB2 (MWU all $P < 0.05$, Supplementary Fig. 2). The 8p11 locus was identified as recurrently amplified as well, wherein, for example, FGFR1, ZNF703 and WHSC1L1 are potential driver candidates. These three genes are indeed significantly higher expressed in the amplified cases (data not shown) precluding isolation of the actual driver from the passenger genes. Tumour suppressors showed lower expression in cases with mutations (CDH1 and PTEN; MWU $P < 0.0001$, Supplementary Fig. 2). For TP53, however, the type of mutation mattered (Supplementary Fig. 3) with obvious deleterious variants such as frameshifts ($n = 19$, MWU $P = 0.002$) and nonsense substitutions ($n = 19$, MWU $P < 0.0001$) clearly showing lower expression, while increased expression of TP53 was found in samples with missense substitutions ($n = 47$, MWU $P < 0.0001$) supportive of a potential tumour promoting role of some of these latter type of TP53 variants[20]. The type of mutation was furthermore associated with copy neutral LOH (Fisher's Exact $P = 0.021$), with cases with missense mutations showing less than expected numbers with copy neutral LOH (17.8 expected, 9 observed), again supportive for a dominant role of missense mutations in the TP53 gene (Supplementary Fig. 3). Finally, we validated earlier reported[1] mutual exclusivity of PIK3CA aberrations, a predominant driver in breast cancer, with any event in PTEN, AKT1 or AKT2 in all 560 cases. Of the 167 cases with a PIK3CA aberration, the majority (91%) lacked any event in the three abovementioned genes ($P = 3.6e-5$ coMEt exact test[21], Supplementary Fig. 4) suggesting these are core constituents in this driver pathway in breast cancer.

As indicated, many of these observations were reported earlier, confirming the validity of our cohort for subsequent analyses. However, a few additional observations were noted for known driver genes; for CCND3, solely found amplified in basal cases, similar expression levels were observed in all but one sample of the amplified cases compared with diploid cases, implying CCND3 is likely not the sole driver in this amplified region (Fig. 2, top left). TCGA results ($n = 960$ cases) confirmed our finding but did show a significant increase in expression levels of CCND3 (ANOVA $P < 0.0001$) in amplified cases supportive of it being an oncogene (Fig. 2, top right). However, with the mean log2 expression level in amplified cases being only 1.2-fold higher compared with diploid cases, this is not the effect one would expect from a biological relevant driver in a particular amplified genomic region. Evaluating genes in the vicinity ($\sim 700$ kb) of CCND3, only USP49 and BYSL could be considered as putative drivers instead, showing a more than 3 fold higher expression in the amplified cases in our cohort. In TCGA data, differences were less pronounced, with BYSL showing the largest fold-change ($1.5 \times$) (Supplementary Fig. 5a).

Next to the above, in our cohort increased GATA3 expression was observed in the 25 cases with GATA3 mutations (MWU $P < 0.0001$, Fig. 2 bottom panel), which may point to a dominant, activating role for mutations in this gene. TCGA results confirmed (ANOVA $P < 0.0001$) this, showing a threefold higher expression found in GATA3 mutated ($n = 95$) compared with wild-type cases ($n = 864$). The vast majority of mutations were found in exon 5 and 6 (22/25 in our cohort and 90/95 in TCGA), but no significant difference was seen in expression levels between

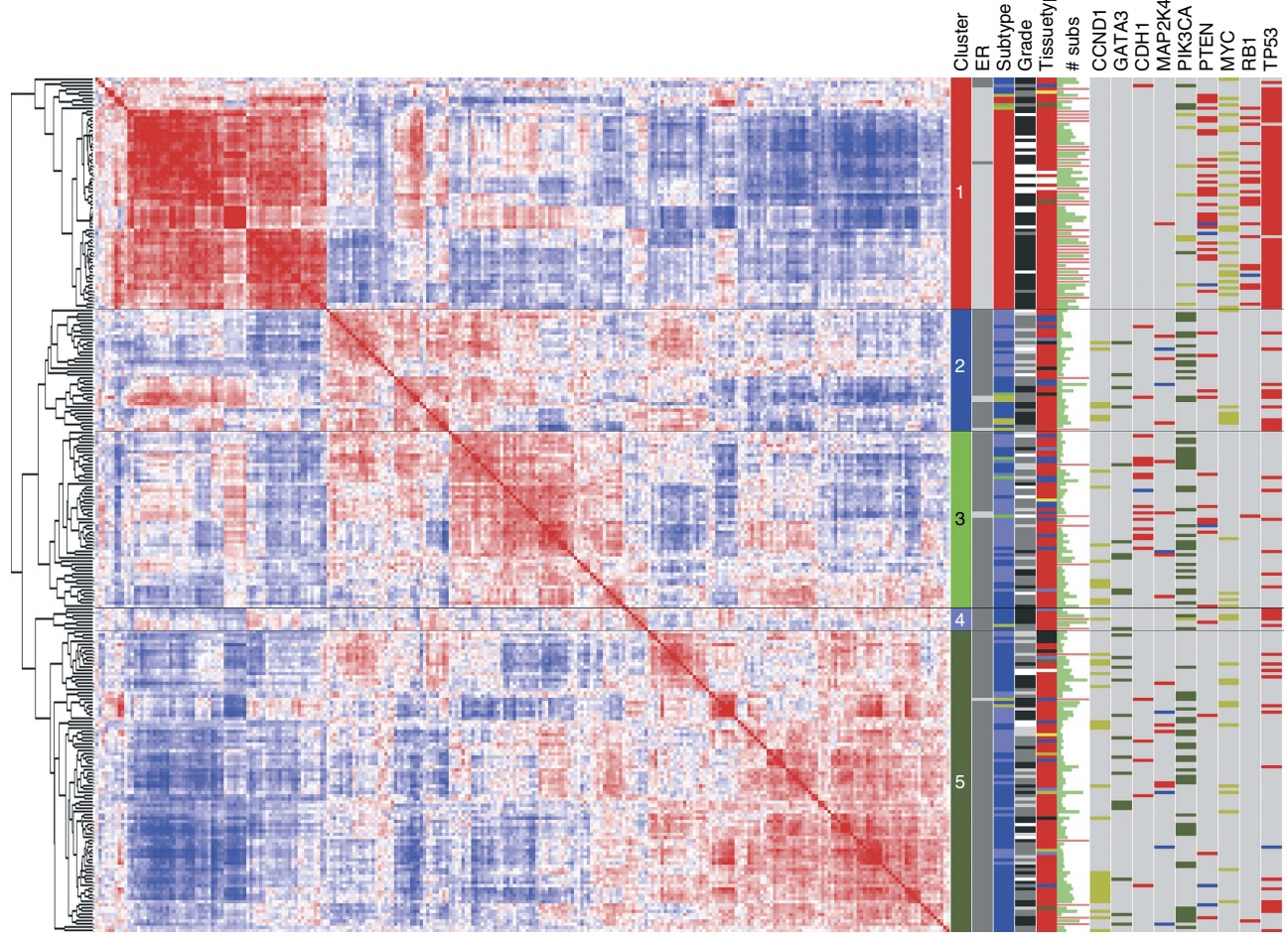

**Figure 1 | Clustered correlation matrix of 266 breast cancer cases.** The left panel shows the dendrogram and clustered correlation matrix (red is positive, blue negative correlation) of 266 breast cancer cases. The top 5,000 most variable transcripts were used for correlating the samples. For the columns in the right panel, colour codes are as follows: ER: ER-positive dark grey, ER-negative light grey. Subtype: Red, basal; dark blue, Luminal B; light blue, Luminal A; green, normal-like; and dark yellow, her2. Grade: white, NA; light grey, grade 1; grey, grade 2; and black, grade 3. Tissuetype: red, ductal; blue, lobular; light blue, micropapillary; grey, mucinous; dark yellow, papillary; yellow, apocrine; and dark green, other type. # subs: The length of the green bar is proportional to the number of substitutions. Cases with >10,000 substitutions are shown with a soft-red coloured bar of equal length. The remaining nine columns show the status of driver genes. Light grey, wild type; dark yellow, copy-number amplification; blue, homozygous deletion; and mutations (substitution, indels, rearrangements) are dark green if activating and red if inactivating.

exon 5 or 6 mutated cases, thus expression changes are independent of the fact that exon 5 mutations predominantly are frameshifts leading to shorter proteins and exon 6 mutations are frameshifts causing proteins with a C-terminal extension (Supplementary Fig. 5b). Noteworthy, the 4% of TCGA cases with an amplification of *GATA3* showed significantly lower expression levels compared with diploid cases (sixfold reduction, ANOVA $P < 0.0001$, data not shown).

**Cell cycle and immune pathways and mutational signatures.** Switching from single genes to studying the effects of the various substitution signatures[9] on the global transcriptome, we first studied the molecular subtypes in relation to the signatures. Significantly higher numbers of signature 3 substitutions but lower number of signature 2 substitutions were found in the basal subtype (KW both $P < 0.0001$) compared with the other subtypes. Substitutions of signature 5 were more abundant in the Her-2 group (KW $P < 0.0001$) while signatures 8 and 13 again showed increased numbers in the basal subtype (KW both $P < 0.0001$). Additional details can be found in Supplementary Fig. 6.

To further evaluate transcriptomic pathways and substitution signatures, we grouped samples according to their dominant mutation signature and compared the cases with, to those lacking that dominant signature (Methods section). The most prominent pathways emerged in ER-positive cases (Supplementary Table 1/ Supplementary Fig. 7) and related to cell cycle, which was significantly associated with the total number of substitutions, while several immune-response pathways were found associated with cases showing predominantly signatures 2 and/or 13. These latter substitution signatures have been accredited[5] to activity of the APOBEC/AID family of cytidine deaminases, targeting the T<u>C</u>N trinucleotide sequence (the underlined base gets mutated). In ER-negatives, several metabolism related pathways were found related with signature 3 substitutions, but these were predominantly driven by few genes or otherwise were always higher expressed in cases with low numbers of that signature (green bars in the geneplot, Supplementary Fig. 7a), giving less confidence that these pathways were of interest. In contrast, the cell cycle pathways consistently showed multiple genes higher expressed in the ER-positive cases with high numbers of the signature (see Supplementary Fig. 7b,c for examples).

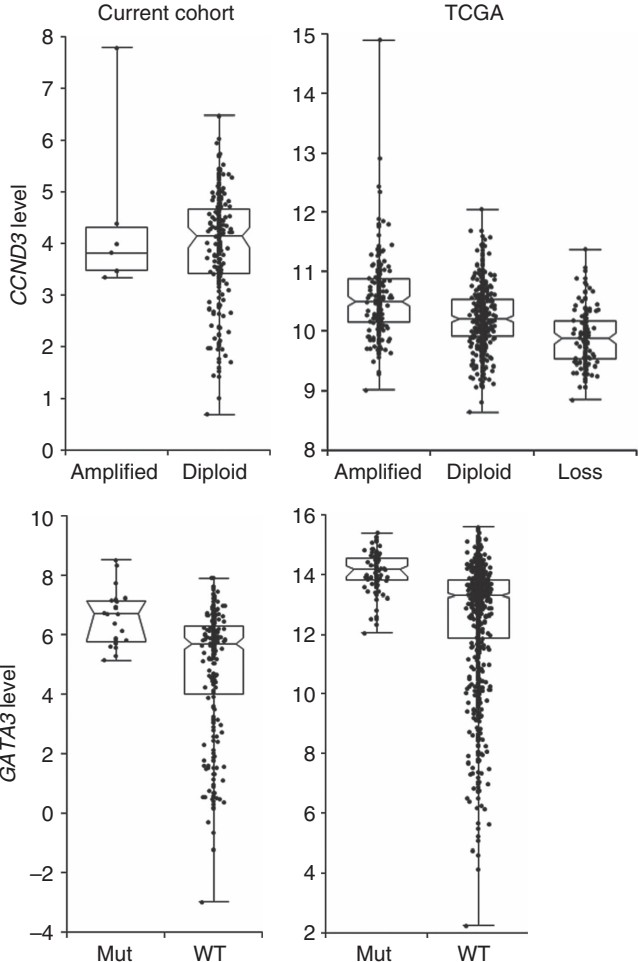

**Figure 2 | Boxplots of CCND3 and GATA3 expression.** All *y* axes show expression levels in our cohort (*n* = 266, left panels, log2-FPKM) and from TCGA cases (*n* = 960, right panels, RNA Seq V2 RSEM, log2) according to copy-number state of *CCND3* (top panel, our cohort *n* = 7 amplified, TCGA *n* = 152 with loss and *n* = 233 amplified) and *GATA3* mutation state (bottom panel, our cohort *n* = 25 mutated, TCGA *n* = 95 mutated).The box is bounded by the first and third quartile with a horizontal line at the median, whiskers extend to the maximum and minimum value. The notch shows the 95% CI of the median.

To validate the pathway results, we studied cell cycle and immune-related gene expression signatures in further detail.

As an objective readout of the cell cycle phenotype, we used genes annotated to the mitotic cell cycle (MCC, GeneOntology:0000278) and associated global expression of these genes to the number and type of substitutions (signatures 6, 17, 20, 26 and 30 were excluded since very few cases harboured these substitution types) and also to pathological grade (Fig. 3a). Next to confirming that the total number of substitutions were higher in the top quartile of samples with high MCC gene expression (MWU $P < 0.0001$), we found that several signatures did similarly, especially signatures 13, 8, 1 and 3 (MWU $P < 0.0001$, $P = 0.0002$, $P = 0.002$ and $P = 0.005$, respectively). Insofar data were available, pathological grade III was also clearly enriched in the top quartile ($\chi^2$ $P < 0.0001$). Thus, irrespective of the type of signatures identified in a patient, a higher number of substitutions was related to increased cell proliferation. Adverse outcome is thus to be expected in patients with a high mutational load, and this was confirmed by evaluating overall and relapse-free survival data (Fig. 3b,c).

The other major theme in the analysis of the transcriptome was the involvement of immune response in relation with particularly the APOBEC-type substitution signatures 2 and 13, also in ER-positive cases. Revisiting the pathway results (Supplementary Fig. 7c) showed a clear association of, for example, *IFNG*, *CTLA4* and several chemokines with these 2 APOBEC-signatures, substantiating a relation with T-cell response. To study objectively if indeed activation of the immune system is mutational signature specific, we related in ER-positive cases (*n* = 192) a previously developed tumour-infiltrating lymphocytes (TIL) RNA expression signature for breast cancer[22] to the diverse mutational signatures, again excluding signatures 6, 17, 20, 26 and 30 due to low numbers (Fig. 4 and Table 1). Samples with overall high expression of the TIL-signature genes (top quartile of cases) had a significantly higher number of signature 13 (MWU $P = 0.0027$) and also of signature 3 (MWU $P = 0.027$). In contrast, the age-related signature 1 (MWU $P = 0.0005$) displayed a significantly lower number in the high TIL group. The number of mutations of the other signature types was not significantly associated with high TIL expression. Next to analysing absolute numbers, we also analysed the data proportionally, by correcting for the total number of mutations, which removes total mutational load as possible confounding factor but this analysis yielded comparable results (Supplementary Table 2).

Even though the number of patients at risk is low, patients with high TIL and low MCC have a significantly better outcome than those with low TIL and high MCC (Fig. 5a,b). We validated this finding in an independent cohort of 625 lymph-node negative, not (neo)adjuvantly hormonal/chemotherapy treated cases (Fig. 5c). The TIL and MCC groups analysed separately also show significant survival differences in this independent cohort (logrank $P = 0.001$ and $P < 0.0001$ for TIL and MCC, respectively. See Supplementary Fig. 8) and when TIL and MCC groups were analysed in a multivariate model, both remained significant (TIL Hazard Ratio 0.53 (95% confidence interval (CI) 0.36–0.79), $P = 0.002$ and MCC HR 1.81 (95% CI 1.33–2.47), $P < 0.0001$).

Next, we investigated the pathological infiltrate status in all ER-positive cases (*n* = 266, which includes those without RNAseq data). We associated the number of mutations per signature with the pathological infiltrate status. Lymphocytic infiltrate was grouped into three groups, having no, mild and moderate-severe infiltrate, and a test for trend across ordered groups showed significantly higher numbers of signatures 3 and 13 (both $P < 0.0001$, see Table 2) in cases with increasing infiltrate, with signature 2 (the other APOBEC signature) showing a weak association as well (test for trend $P = 0.022$). Of note, the results for signature 3 may be hampered by the fact that many cases lack signature 3 mutations entirely; of the *n* = 266 cases in this analysis 230 (86%) have zero signature 3 mutations. An analysis with the proportion of substitutions of a signature yielded largely similar results (Supplementary Table 3), in the sense that signatures 3 and 13 remain strongly associated with the infiltrate status (test for trend $P < 0.0001$), whereas signature 1 and 5 showed higher proportions in cases with decreasing infiltrate (test for trend $P = 0.015$ and $P < 0.0001$, respectively). To exclude that individual drivers themselves were not responsible for the observed association we investigated if mutation (including substitution, indel, rearrangement, copy-number variation) of the driver genes was directly associated with infiltrate levels. The following genes were analysed: *ARID1A, CCND1, CDH1, GATA3, MAP2K4, MAP3K1, MLL3, MYC, PIK3CA, PTEN, TP53* and *ZNF217* (*n* = 10, 32, 17, 22, 17, 20, 11, 21, 64, 13, 33 and 11, respectively), but none of the investigated genes was found to be significantly associated with infiltrate status (Fisher's Exact test, $P > 0.05$).

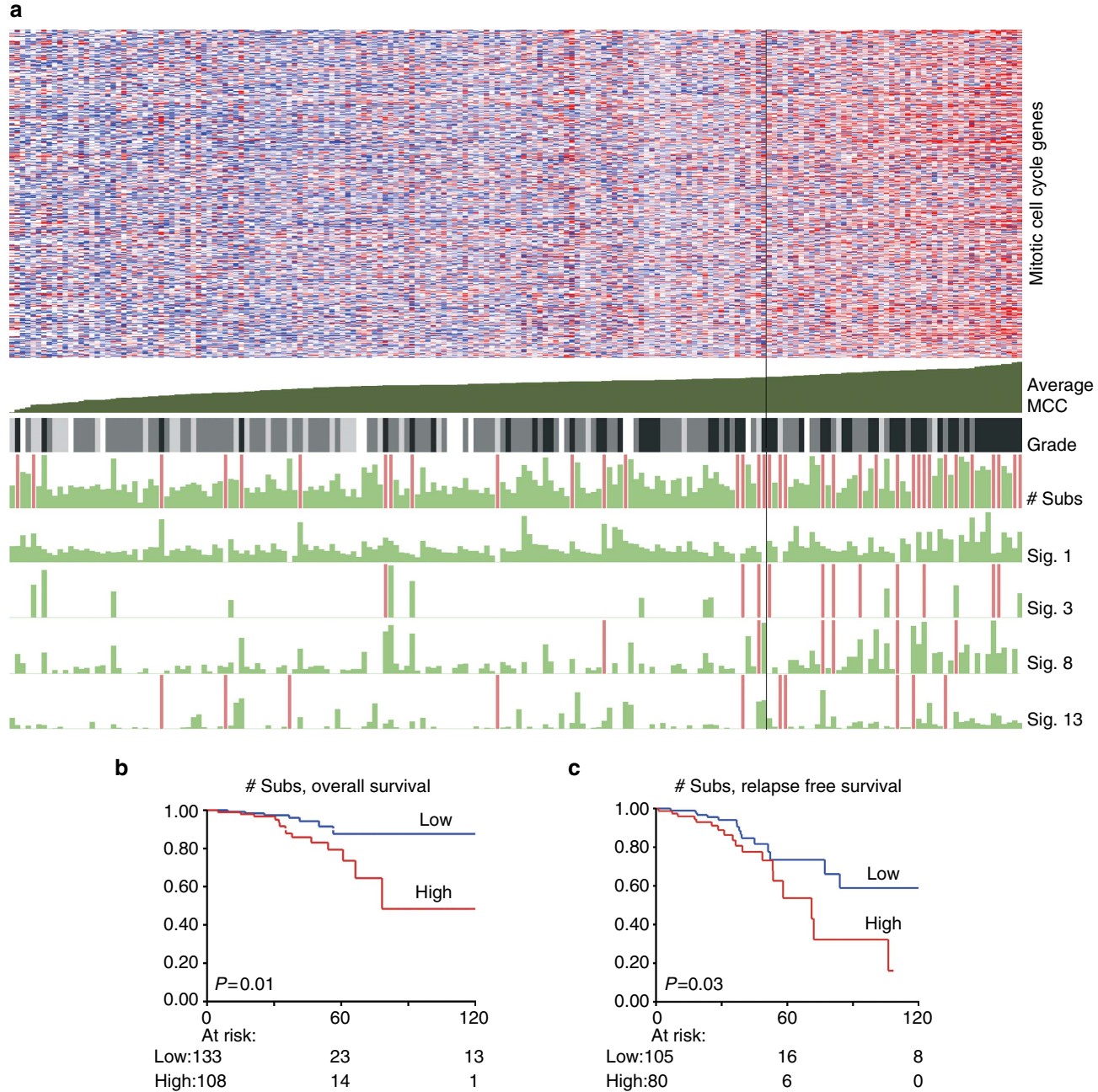

**Figure 3 | Mitotic cell cycle gene activity related to mutational signatures and outcome. (a)** The average expression of genes ($n = 409$) from Gene-Ontology term Mitotic Cell Cycle (GO:0000278) were used to rank ER-positive samples. The vertical black line indicates the third quartile border. Top panel: heatmap of median centred expression values in log2-FPKM, red indicates above median, blue below median expression. Genes are in rows, samples in columns. Below the heatmap: 'Average MCC' shows the average expression of the cell cycle genes. Grade: pathological grade; white, NA; light grey, grade 1; grey, grade 2; and black, grade 3. Last five rows: the length of the green bar is proportional to the number of substitutions. # subs: the total number of substitutions, samples with >10,000 substitutions are shown with a soft-red coloured bar and are of equal length. For the columns labelled Sig. 1, 3, 8 and 13, soft-red indicate samples >3,000 of such substitutions. **(b,c)** Overall and relapse-free survival Kaplan–Meier curves. Blue indicates patients with less than the median number of substitutions, red indicates higher than median. $P$ values are logrank-test values. The x-axis shows time in months, y-axis shows the proportion of surviving patients.

**Amino acid properties and immune response.** To try to understand why only certain mutation signatures relate to a more effective immune response, we determined *in-silico* which substitutions yield predicted neo-epitopes[23,24] but found no striking signature specific difference in the fraction of predicted neo-epitope presenting substitutions (Supplementary Fig. 9a). We also studied the altered chemical properties[25,26] of the mutational signature induced amino acid changes. However, changes in hydrophobicity (Supplementary Fig. 9b) were not strongly differently proportioned across the substitution signatures. In contrast, amino acid substitutions resulting from particularly signature 13 but also from signature 2 displayed a clear increase in charge (Supplementary Fig. 9c), while substitutions caused by signature 1—which were associated with lower infiltrate—displayed a loss of a charge instead. The number of signature 13 substitutions that lead to an increase in charge was

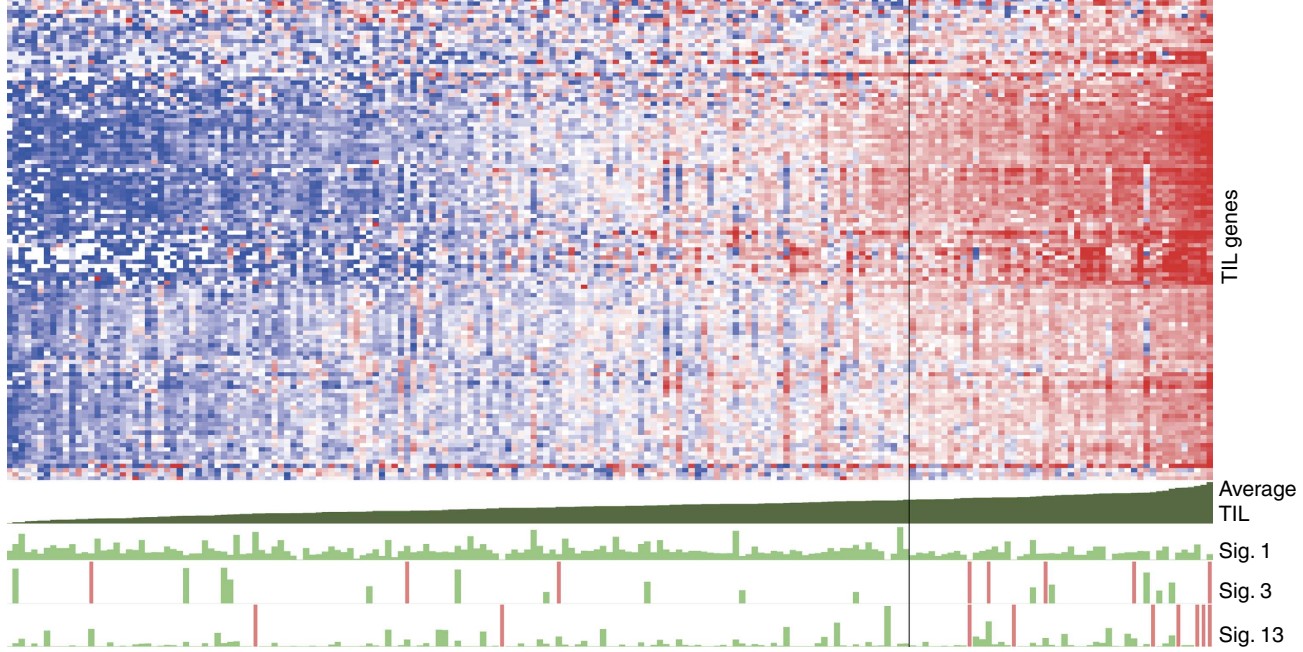

**Figure 4 | Activity of TIL-signature genes related to mutational signatures.** The average expression of genes ($n = 116$) from a TIL specific RNA-signature was used to rank ER-positive samples. The vertical black line indicates the third quartile border. Top panel: heatmap of median centred expression values in log2-FPKM, red indicates above median, blue below median expression. Genes are in rows, samples in columns. Below the heatmap: the first row shows the average expression of the TIL genes. Last three rows: the length of the green bar is proportional to the number of substitutions of the indicated signatures. Samples with $> 3,000$ substitutions are shown with a soft-red coloured bar and are of equal length.

| Signature | n | MWU* | Average high TIL | Average low TIL |
|---|---|---|---|---|
| 1 | 183 | 0.0005 | 532.8 | 752.5 |
| 2 | 179 | 0.64 | 2,313.1 | 328 |
| 3 | 24 | 0.027 | 581.4 | 209.5 |
| 5 | 166 | 0.27 | 1,033 | 1,022.2 |
| 6 | 2 | N.D. | 17,791 | 0 |
| 8 | 123 | 0.59 | 663.6 | 453.3 |
| 13 | 115 | 0.0027 | 1,897 | 309.5 |
| 17 | 6 | N.D. | 9.7 | 23 |
| 18 | 32 | 0.39 | 37.8 | 64.4 |
| 20 | 1 | N.D. | 0 | 0 |
| 26 | 3 | N.D. | 171.5 | 87.6 |
| 30 | 1 | N.D. | 0 | 0 |

**Table 1 | Association of the number of substitutions per signature by TIL-signature group.**

N denotes the number of samples with $> 0$ of a particular substitution signature. The average number of substitutions is listed per TIL-signature group; high TIL indicates the top quartile of samples, low TIL the rest of the samples.
*P value of MWU.

significantly associated with increasing lymphocytic infiltrate (test for trend $P < 0.0001$); a similar test for signature 2 substitutions showed a $P$ value of 0.014. Analysing the total number of substitutions leading to increased electric charge, irrespective of signature type, also showed increasing numbers with increasing lymphocytic infiltrate (test for trend $P < 0.0001$) which remained significant even when signature 13 substitutions were excluded (test for trend $P < 0.0001$).

## Discussion

In this paper we investigated the mutation signatures present in primary breast cancer and their relation to the transcriptome. Although somewhat impeded by the fact that most tumours display a multiplicity of signatures, we were able to show that

individual signatures have specific effects on the gene expression phenotype of tumour cells. As may be noted from the results, the main observations were obtained in ER-positive cases. The lack of results in ER-negative cases is attributed to the more homogeneous nature of this subtype; for example, virtually all ER-negative patients have high infiltrate levels (in our cohort only four cases display a nill infiltrate phenotype) and a high mutational load (just two ER-negative cases are among the bottom 20% after ranking on mutational load), which impedes identifying expression differences due to the lack of a sizeable contrast group.

Regarding the global transcriptome in relation to the substitution signatures, first, we observed that the total number of substitutions was positively associated with expression of cell cycle genes, independent of which of the individual substitution signatures contributed to the total. Reviewing the driver genes, as expected *CCND1* and *MYC* amplified cases, as well as *TP53* mutated cases were enriched in the top quartile MCC expression (Fisher's Exact test, all $P < 0.0001$). The relation between an active cell cycle and various known drivers of cell cycle progression (*CCND1, MYC, TP53*) and poor outcome in ER-positive patients has been widely investigated[16,27–31], thus our observation that the mutational burden (defined here as the total number of substitutions, regardless of type) was also associated with shorter DFS and OS may be not surprising. The association of mutational burden and poor outcome in ER-positive patients was described earlier as well[32], but using data based on exome sequencing, which we firmly validated here using whole genome sequencing data, with the latter enabling removal of the possible bias which may exist on the total mutational burden derived from the transcribed part (exome) compared with the entire genome (WGS). Noteworthy is that both mutational signatures which are replication dependent (for example, signature 13) and those that are not (for example, signature 8)[9] contribute to the mutational burden. This suggests that mutational load as a result of

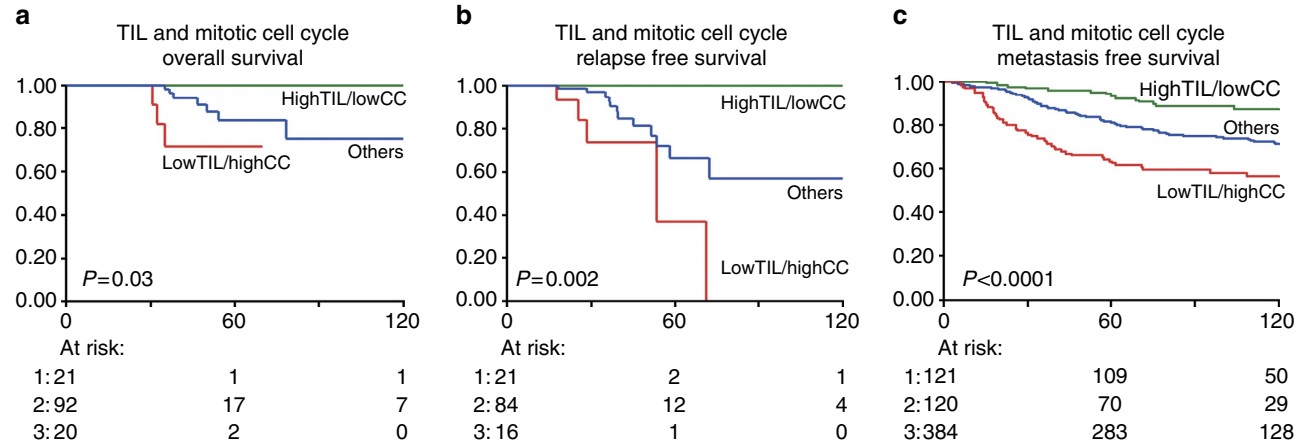

**Figure 5 | Combined MCC and TIL-signature genes and outcome.** Overall (**a**) and relapse-free (**b**) survival Kaplan − Meier curves of our cohort and (**c**) metastasis-free survival of independent in-house and public data sets. Green line indicates patients with high expression of TIL genes (top quartile) and low expression of MCC genes (bottom three quartiles). The red line indicates patients with low expression of TIL genes and high expression of MCC genes. Blue indicates the remaining patients. P values are logrank-test for trend values. The x-axis shows time in months, y-axis shows the proportion of patients. For the numbers at risk, 1 indicates the highTIL/lowCC group (green line), 3 indicates the lowTIL/highCC (red line) and 2 the remaining patients.

**Table 2 | Association of the number of substitutions per signature by lymphocytic infiltrate group.**

| Signature | N | Cuzick* | Average nill | Average mild | Average moderate/severe |
|---|---|---|---|---|---|
| 1 | 253 | 0.537 | 592.9 | 728.5 | 649.9 |
| 2 | 248 | 0.022 | 169.5 | 657.7 | 1,401.5 |
| 3 | 36 | <0.0001 | 127.7 | 197.8 | 944.2 |
| 5 | 237 | 0.158 | 1,131.2 | 1,040.7 | 1,059.4 |
| 6 | 2 | N.D. | 0 | 54.9 | 355.8 |
| 8 | 172 | 0.117 | 322.4 | 435 | 785.4 |
| 13 | 174 | <0.0001 | 91.5 | 573.1 | 1,613.9 |
| 17 | 9 | N.D. | 11.7 | 20.9 | 14.5 |
| 18 | 45 | 0.935 | 32.6 | 90.3 | 34.2 |
| 20 | 0 | N.D. | 0 | 0 | 0 |
| 26 | 3 | N.D. | 0 | 126 | 164.6 |
| 30 | 0 | N.D. | 0 | 0 | 0 |

N denotes the number of samples with >0 of a particular substitution signature. The average number of substitutions is listed per lymphocytic infiltrate group.
*P value of Cuzick's nonparametric test for trend across ordered groups.

insufficient or improper repair during DNA replication, as well as those from other sources, both contribute to more rapid progression of ER-positive disease.

Second and interestingly, we observed a significant, positive relation between substitutions of signatures 13 and 3 and a TIL gene expression signature and also to increasing levels of TILs. The association of signature 3 substitutions with immune-response pathways is not readily explained by the type of substitutions—which are broadly patterned—or by the investigated amino acid properties; we speculate that the strong association of signature 3 with inactivated *BRCA1/2* (ref. 9) could make this signature a proxy for BRCA-ness (with its particular rearrangements) in ER-positive cases. Inactivated BRCA is tightly linked with loss of homologous recombination (HR) repair mechanisms, leading to distinct rearrangement patterns[9]. We speculate that these rearrangements, which can comprise up to hundreds of events in a sample, leading to randomly aberrant proteins, potentially triggering an immune response. And, although the exact *modus operandi* of how signature 3 or BRCA-ness could increase immune response is as of yet obscure, higher lymphocytic infiltrate have been previously reported in familial *BRCA1* (ref. 33) and *BRCA2*-affected tumours[34], the

former study also showing significantly better relapse-free survival for BRCA-patients with higher infiltrate levels.

The connection between TIL and signature 13 substitutions may be based on the distinguishing feature of signature 13; its clear APOBEC pattern[2,5], deaminating cytidines in a TCN context, favouring C to G substitutions. We hypothesize that TILs are enriched at sites where the proper peptides are presented; a possible way signature 13 substitutions could activate this immune-response is via its resulting neo-epitopes. One difference we observed for altered peptide sequences resulting from signature 13 substitutions as compared with other equally abundant ones originating from other sources (for example, signature 1, 5, 8 substitutions) was not the number of predicted neo-epitopes, but rather its tendency to being positively charged. The observed increase in charge in signatures 2 and 13 can be readily explained by the fact that these signatures frequently affect the only two negatively charged amino acids Glu and Asp, respectively, which are modified in this mutational context into Lys (signature 2) and His (signature 13), both positively charged. Thus, these latter 2 signatures are most efficient in generating mutated peptides with increased electric charge. Interestingly, patients with an increased lymphocytic infiltrate showed, irrespective of signature type, higher numbers of positively charged amino acid substitutions, indicating that the association of signature 13 with TIL might be because positively charged amino acids are more commonly generated in this signature type.

Concluding, we show that ER-positive breast cancer responds to DNA mutations with two specific changes in the transcriptome. On the one hand, the cell cycle with its associated adverse clinical outcome appears to be more active with an increasing number of mutations, regardless of their underlying signatures. On the other hand, we observe a signature specific association with immune response, possibly arising from an increased number of neo-antigens that consequently may stimulate the immune response more effectively. However, independent confirmation of our finding is needed as well as mechanistic studies explaining the observed association. Still, our results augment earlier reports[35–37], which postulate that a high mutational load is sufficient for the immune system to sense one or more neo-epitopes as non-self. However, by untangling the mutational load into specific signatures, our results suggest that the breast tumour cell, while gaining advantage by those mutations stimulating proliferation, may inadvertently provoke

the immune system more effectively if the mutational load is moulded through specific mutational processes. Exploiting this knowledge by purposefully augmenting T-cell reactivity against amino acid substitutions resulting from the proper DNA substitution type could improve cancer immunotherapies.

## Methods

**Cohorts.** The sample cohort used in the study has been described in detail recently[9]. To recapitulate, DNA and RNA of breast cancer tumours were analysed using WGS (WG-DNA data), copy-number analysis, methylation profiling, RNA sequencing and miRNA analysis. Here, we studied all patients with available RNAseq and WG-DNA data ($n = 266$, no replicates). Sequencing protocols, QC and post-processing of data as well as the DNA substitution and rearrangement signatures and data availability are described elsewhere[9]. For the RNA-sequence cohort, IHC-scored Her2-positivity (as ascertained by a panel of pathologists according to the current standard practice) was an exclusion criterion, though at the time of analysis, some cases showed DNA amplification of the Her2 locus. Gene expression values were available as log2-FPKM values for 49,738 transcripts. For the correlation matrix (Fig. 1) transcripts with $> 20\%$ missing data were excluded, and the top 5,000 variable transcripts—which included besides 3,484 protein-coding genes, 1,516 non-coding RNA species such as lincRNAs, snRNAs and snoRNAs and so on—were median centred and used to correlate (Pearson) all samples with each other. The resulting correlation matrix was hierarchically clustered[38] (Cluster 3.0) using uncentered correlation as distance metric. The molecular subtypes were established using the AIMS method[6].

**Pathway analysis.** Since the transcriptome is heavily driven by ER-status, pathway analyses were performed within ER-positive and -negative breast cancer cases separately. In the pathway analysis each signature was analysed separately: the proportion of substitutions of a particular signature was used for grouping samples. Samples which had at least 50% of a particular signature were compared with samples having $< 20\%$ of such substitutions, where both groups should have at least 15 cases. In ER-positive cases, signature 5 ($n = 54$ $> 50\%$ proportion versus $n = 32$ $< 20\%$) and the APOBEC-signatures (Sig 2 + 13 combined) with $n = 18$ $> 50\%$ proportion versus $n = 45$ $< 20\%$) were available for analysis and in ER-negative cases, signature 3 ($n = 24$ $> 50\%$ versus $n = 22$ $< 20\%$) could be evaluated. In addition, disregarding the type of signature, the total number of substitutions was used to group samples with the most extreme number of substitutions (top and bottom 20%). Initially samples were grouped in a low and high mutational burden group, separately for ER-positive and -negative cases. However, in the ER-negatives, samples in the bottom 20% still had a high absolute number of substitutions. Since our aim was to identify differential pathways associated with the absolute tumour burden, we wanted to have a bottom 20% group with absolute low numbers of substitutions. Samples were thus ranked according to mutational burden independent of ER-status to obtain the bottom and top 20% groups. In ER-positive cases, this gave a group of $n = 15$ in the top 20% versus $n = 52$ in the bottom 20%. In ER-negative cases there were only two samples in the bottom 20%, precluding a meaningful analysis. Pathway analysis were performed using the R-package 'global test'[39] using Biocarta (www.biocarta.com) and KEGG (ref. 40) as annotation databases. All $P$ values were corrected for multiple testing (Bonferroni–Holm) and checked by re-sampling 1,000 times to ascertain the number of times an equally sized, randomly chosen group of genes is at least as significant as the true set of genes belonging to a pathway. Pathways were considered of interest if the $P$ value of the global test after correcting for multiple testing and the re-sampling $P$ value were both below 0.05. Second, pathways were considered relevant pathways if these consistently showed multiple genes higher expressed in the cases with high numbers of a signature.

**Neo-antigen prediction and hydrophobicity of amino acids.** To ascertain which of the amino acid changing substitutions potentially contain a neo-epitope, all unique, non-silent, non-terminating amino acid substitutions were selected ($n = 26,475$) and we focused our analyses on CD8 T-cell epitopes. FASTA files of the wild-type sequences were obtained via the consensus coding sequence database (http://www.ncbi.nlm.nih.gov/CCDS/CcdsBrowse.cgi) and custom Perl scripts were used to mutate the wild-type sequence according to the identified substitution in our cohort. Since the software to evaluate neo-epitopes uses all possible 9-mer peptides in a sequence, we selected a 17-mer sequence with the mutated amino acid in the middle – thus making sure the mutated amino acid was always present in all possible 9mers of that sequence. Care was taken if the substitution was too near to the start or end of a gene to ensure a 9mer was always present. The mutated FASTA sequences were uploaded to NETMHC v3.4 (refs 23–24) (http://www.cbs.dtu.dk/services/NetMHC/ ) to predict the binding affinity to the following HLA-A2 alleles, representing the most prevalent MHC class I alleles among the Caucasian population: HLA-A2:01, 02, 03, 06, 11, 12, 16, 17, 19 and 50. With a possible eight sequences to be tested per gene, this gives a matrix of at most 80 predictions. As defined[23], a predicted EC50 $< 50$ nM of a peptide towards HLA-A2 alleles indicates a 'strong binder' and if any of the possible predictions was found $< 50$ nM the gene was considered harbouring a potential neo-antigen.

Changes in hydrophobicity of mutated amino acids was ascertained using the scale of Kyte & Doolittle[41] where the level of the mutated amino acid was subtracted from the wild-type amino acid. A positive difference ($> 0$) was labelled as 'increase' and $< 0$ as 'decrease' in hydrophobicity. Changes in electric charge were evaluated by assigning Aspartic acid and Glutamic acid as negatively charged and Lysine, Arginine and Histidine as positive. Next, base substitutions were evaluated if a resulting amino acid change led to a change in electrical charge. Since all substitutions are assigned to a mutational signature, the total number of charge changing (and hydrophobicity changing) substitutions per signature was established.

**Independent validation set.** In-house and publicly available gene expression data of lymph-node negative primary breast cancer patients, who were not treated (neo)adjuvantly with hormonal/chemotherapy and available metastasis-free survival data were used, leading to a cohort of 867 patients ($n = 625$ ER-positive). This selection enables a pure study on prognosis. Data were gathered from Gene Expression Omnibus (http://www.ncbi.nlm.nih.gov/geo/) entries GSE2034, GSE5327, GSE2990, GSE7390 and GSE11121, with all data available on Affymetrix U133A chip. Raw.cel files were processed using fRMA (ref. 42) parameters (median polish), after which batch effects were corrected using ComBat (ref. 43). The average expression of TIL genes[22] and MCC genes (Gene Ontology:0000278) were calculated of each sample, and the samples in the top quartile for TIL and the top quartile for MCC were used to group the patients in three groups; those with high TIL/low MCC, those with low TIL/high MCC and the remaining patients.

**Statistics.** Associations between gene expression levels and gene-status (mutant, amplification and so on) were analysed using MWU, or KW when more than two categories were evaluated. In the analyses where the expression level of a gene was associated with its own status and not with other genes, no multiple testing correction was necessary. To test for a trend across ordered groups (for example, number of substitutions in increasing grade), Cuzick's nonparametric test for trend was used, an adjunct to the KW test. Pearson's $\chi^2$ test was used to evaluate categories (for example, mutation status versus molecular subtype) or by Fisher's Exact test when number of events were low. Stata v13 (StataCorp, Texas, USA) was used for the statistical tests and Kaplan–Meier survival curves. A two-sided $P$ value of $< 0.05$ was considered statistically significant.

**Data availability.** WGS, RNA-seq and methylation data that support the findings in this study are available at the European-Genome Phenome Archive (https://www.ebi.ac.uk/ega) under the accession code EGAS00001001178. For the validation data, in-house GSE2034 and GSE5327, and otherwise public data GSE2990, GSE7390 and GSE11121 are available at the Gene Expression Omnibus website (http://www.ncbi.nlm.nih.gov/geo/). All other data are contained within the Article and its Supplementary files or available from the authors on request.

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

## Acknowledgements

This work has been funded through the ICGC Breast Cancer Working group by the Breast Cancer Somatic Genetics Study (a European research project funded by the European Community's Seventh Framework Programme (FP7/2010-2014) under the grant agreement number 242006); the Triple Negative project funded by the Wellcome Trust (grant reference 077012/Z/05/Z).

Personally funded by grants: FGR-G, SM, KR, SM were funded by BASIS. JAF was funded through an ERC Advanced Grant (ERC-2012-AdG-322737). SN-Z is a Wellcome Beit Fellow and personally funded by a Wellcome Trust Intermediate Fellowship (WT100183MA). ALR is partially supported by the Dana-Farber/Harvard Cancer Center SPORE in Breast Cancer (NIH/NCI 5 P50 CA168504-02). AS was supported by Cancer Genomics Netherlands (CGC.nl) through a grant from the Netherlands Organisation of Scientific research (NWO). MS was supported by the EU-FP7-DDR response project. CD was supported by a grant from the Breast Cancer Research Foundation. EB was funded by EMBL. JE was funded by The Icelandic Centre for Research (RANNIS).

For technical or administrative support: Lira Mamanova from the Wellcome Trust Sanger Institute is thanked for her expertise in RNA library preparation. We thank contributors Luc Dirix (Antwerp, Belgium) and Suet-Feung Chin (Cambridge, UK), Jon G Jonasson (University of Iceland), The Icelandic Cancer Registry and from Nijmegen (The Netherlands): Margrete Schlooz-Vries, Jolien Tol, Hanneke van Laarhoven, Fred Sweep and Peter Bult. Benita Kiat Tee Tan from the National Cancer Centre Singapore and Singapore General Hospital is thanked for contributing patient samples for this study as are OSBREAC, the Oslo Breast Cancer Research Consortium, Norway (http://ous-research.no/home/kgjebsen/home/14105), the Brisbane Breast Bank, Australia and the Tayside Tissue Bank, Dundee, UK.

Finally, we acknowledge all members of the ICGC Breast Cancer Working Group.

## Author contributions

M. Smid and J.W.M.M. wrote the main paper. M. Stratton, S.M., S.N.-Z, H.G.S., J.A.F., J.W.M.M. were involved in the strategy and supervision of the project. Experiments were performed by F.G.R.-G, A.M.S., W.J.C.P.-V.d.S., M.v.d.V.-D., A.v.G., J.S., A.B.B., K.B., M.R. and A.L. M. Smid, S.N.-Z, J.S., A.B.B., M.J.v.d.V, A.L.R., A.B., H.D.D., R.D., M.E.M.-v.G., C.H.M.v.D., S.R.L., M.R., A.V., A.B.-D., E.B. and J.W.M.M. analysed data. Samples were contributed by J.A.F., J.W.M.M., A.L.R., A.F., C.P., A.M.T., C.C., P.N.S., P.T.S., S.R.L., S.V.L., C.D., S.T., O.A.S., A.B., A.V.-S., P.A.F., S.K., T.K., G.T., A.V., A.L., A.-L.B.-D., G.M., R.S., G.G.G.M.V.d.E.

## Additional information

**Competing financial interests:** The authors declare no competing financial interests.

