## [Peer Review File · Nature Communications]

Reviewers' comments:

Reviewer #2 (Remarks to the Author):

See earlier reviews which we think transferred from another journal.

In the revised submission, the authors have added additional statistical support for their discovery that specific types of mutations can induce an effective immune response associated with good prognosis in breast cancer. They have also added text to the main body and the discussion putting their work in the context of previously published results in breast and other cancers. As such, even without functional experiments to test the presented hypotheses, this work is worthy of publication assuming all statistical analyses have been reviewed and deemed appropriate.

Reviewer #3 (Remarks to the Author):

This paper strives to apply the mutational signature approach to categorizing the mutations found in malignancies to the biological/clinical behavior of the tumors. The original establishment of signatures seems to me to be a much more straightforward endeavor since one expects that particular mutational mechanisms would be involved in particular tumors, and that those would produce specific sets of mutations. The signature system that is being used is a reasonable choice-categorizing mutations by looking at the nucleotides on either side of the substitution, and then grouping each of these "basis" mutations into signatures using data from lots of tumors. But that is only one of an infinite number of sets of "basis" mutations that one could use. For example one could only use the nucleotide on one side of the substitution or the other, or use two nucleotides on one or both sides of the substitution etc.

It seems to me that the interesting scientific question is how much useful information is summarized/simplified by using this set of signatures, which were defined in earlier publications from many of the same investigators. The fact that signatures were found means that the expectation about mutational mechanisms was validated. The present paper has the opportunity to address the issue of the broader utility of the signature approach. However instead of providing information on this larger question they highlight the one positive thing that they did find-a possible link of some signatures to immune status of the tumors, see below.

It seems to me that a key statement in the paper is the sentence in the Discussion that says that their work was somewhat impeded by the fact "most tumors display a multiplicity of signatures". It is as if the biology was trying to add up these signatures to reconstitute the individual "basis" mutations that went into the initial analysis-eg the signatures are of limited biological relevance to gene expression. That certainly is the case with the cell cycle expression, where total number of mutations and not anything with signatures was the significant correlate. Among the questions that come to mind and do not seem to be addressed in the paper are: a) whether there is any relation of the mutational signatures to breast cancer subtype as defined by the expression analysis, and b) if one forgets about mutation signatures and focuses instead on the individual "basis" mutations, is there any association of any of them with immune status? The latter question might have a positive answer if there is a single recurrent mutation that has a large effect on immune status.

It seems to me that Figure 5 is in some sense largely independent of any analysis of mutation signatures. It shows the survival based on expression measurements, with the TIL expression determined from the lymphocyte infiltrate mixed with the tumors. Moreover, parts A and B are of doubtful utility since only one or two patients are being followed in the key groups 1 and 3. A single event could have totally changed the picture. Looking at the data of Figure 4 it appears that only a small proportion of the high TIL tumors have a large presence of the signatures 3 and 13. Thus while these signatures may be significantly correlated with high TILs, the presence of signature of the signature is unlikely to provide much information about outcome. It might be useful for the authors to present that analysis, assuming that they have the follow up that will permit it.

Reviewer #4 (Remarks to the Author):

Using bioinformatics the study compares WGS data (from Nik-Zainal et al Nature 2016) and RNA-seq data for 266 breast cancer cases.

It is schematically organized in three parts.

The first part examines the expression of mutated genes to help define "driverness". Nothing much is novel but, being based on state-of-the-art methods, it should be at least comprehensive.

1 - Because emphasis seems to be made on luminal cases I am surprised that nothing is said about 8p12 amplified genes, such as FGFR1, WHSC1L1 and ZNF703.

2 - The case of GATA3 is interesting. Functional validation of mutated GATA3, such as overexpression, ChIP-seq..., would have been informative. Supervised analyses of mutated vs non-mutated cases could have been done.

3 - Please include a table with P53 mutations separating the two potential types, expressed/oncogenic vs absent/TSG. Relate this to 17p LOH.

The second part is about proliferation.

4 - It would be interesting to relate proliferation/mitotic cycle with mutations/amplifications/deletions, and to signatures. For examples in Fig 3, is there a gradient of MYC, CCND1... amplifications (more at right)? It would be interesting to add ER- cases.

The third part is about mutational burden, mutational signatures and immune response. In the absence of further validation studies on mutated/amplified/deleted genes, and given the recent availability of immune-based treatments, this part is actually the most (only?) interesting.

5 - What is the importance of mutational burden rather than mutational type is not clear in the

written text. This point should be made clearer.

6 - The analysis should not have been limited to ER+ cases. This is really surprising because immune signatures have been shown to be important in basal/triple negative/medullary cases (e.g. Teschendorff et al., LCK metagene...). There is much to do here. E.g. samples classified upon these expression signatures may be examined for mutational signatures.

7 - There is a way to establish a BRCAness score (Popova et al., 2012). Please test it and see if it matches mut S3. Mut S3 seems to be associated with ER-, which BRCA1 are. Again, test this.

8 - Concerning prognosis, were multivariate analyses done?

In conclusion, the study is timely and potentially interesting but as such, is a bit superficial.

Reviewers' comments:

Reviewer #2 (Remarks to the Author):

See earlier reviews which we think transferred from another journal.

In the revised submission, the authors have added additional statistical support for their discovery that specific types of mutations can induce an effective immune response associated with good prognosis in breast cancer. They have also added text to the main body and the discussion putting their work in the context of previously published results in breast and other cancers. As such, even without functional experiments to test the presented hypotheses, this work is worthy of publication assuming all statistical analyses have been reviewed and deemed appropriate.

***Response:** we thank the reviewer for this favourable response and appreciate that he/she considers our work in the current format suitable for publication.*

Reviewer #3 (Remarks to the Author):

This paper strives to apply the mutational signature approach to categorizing the mutations found in malignancies to the biological/clinical behavior of the tumors. The original establishment of signatures seems to me to be a much more straightforward endeavor since one expects that particular mutational mechanisms would be involved in particular tumors, and that those would produce specific sets of mutations. The signature system that is being used is a reasonable choice-categorizing mutations by looking at the nucleotides on either side of the substitution, and then grouping each of these "basis" mutations into signatures using data from lots of tumors. But that is only one of an infinite number of sets of "basis" mutations that one could use. For example one could only use the nucleotide on one side of the substitution or the other, or use two nucleotides on one or both sides of the substitution etc.

It seems to me that the interesting scientific question is how much useful information is summarized/simplified by using this set of signatures, which were defined in earlier publications from many of the same investigators. The fact that signatures were found means that the expectation about mutational mechanisms was validated. The present paper has the opportunity to address the issue of the broader utility of the signature approach. However instead of providing information on this larger question they highlight the one positive thing that they did find-a possible link of some signatures to immune status of the tumors, see below.

***Response:** We agree with the reviewer that indeed the aim of our study was to relate biology present in the transcriptome to recurrent mutations and particularly to recurrent mutational signatures identified in breast cancer. We used the 'nucleotide to each side' approach, as it is referred to by this reviewer, since there is consensus in the literature to the existence of recurrent somatic mutational patterns therein (see <http://cancer.sanger.ac.uk/cosmic/signatures>). Furthermore, for several of these recurrent patterns a clear intrinsic biological (e.g. MSI), or extrinsic biochemical rationale (UV exposure, smoking etc.) is*

connected to them. Furthermore, in the whole genome data, 12 different recurrent patterns were validated and/or identified allowing us to address the question of transcriptomic effects in relation to these signatures for the first time. We appreciate that the reviewer considers our approach to connect biology to these recurrent patterns a reasonable choice. We also do agree with the reviewer that there are many other possible ways to categorise recurrent somatic variants but for now other ways to meaningfully group them have not been described and it was not our aim to do so.

However, we do not fully agree with the reviewer that we focused on just one issue. In this study we did address the broader utility by associating all the identified mutational signatures to the whole transcriptome in a pathway analysis. Thus we did explore in an unbiased manner which of the transcriptome pathways were differently expressed between cases having a particular signature versus those who did not. Only two major pathways stood out and these were further explored. So we did not pick out one pathway which we deemed interesting, we merely continued with pathways which were significantly connected to one or more of the 12 mutational patterns present in the current dataset.

It seems to me that a key statement in the paper is the sentence in the Discussion that says that their work was somewhat impeded by the fact "most tumors display a multiplicity of signatures". It is as if the biology was trying to add up these signatures to reconstitute the individual "basis" mutations that went into the initial analysis-eg the signatures are of limited biological relevance to gene expression. That certainly is the case with the cell cycle expression, where total number of mutations and not anything with signatures was the significant correlate.

Response: *It is correct that multiple tumours have a multiplicity of mutation signatures and that biology related to them may be intermingled. Therefore, in our initial discovery phase (supplemental information on pathway analysis) we did focus on specimen particularly enriched for somatic variants representing only one particular signature. With regard to the cell cycle pathway; in our view for the cell cycle to be activated it does not seem to matter from which signature it arises, it just matter that they arise. Thus, the sheer number of mutations makes the cancers more likely to have their proliferation pathways turned on.*

Among the questions that come to mind and do not seem to be addressed in the paper are: a) whether there is any relation of the mutational signatures to breast cancer subtype as defined by the expression analysis, and b) if one forgets about mutation signatures and focuses instead on the individual "basis" mutations, is there any association of any of them with immune status? The latter question might have a positive answer if there is a single recurrent mutation that has a large effect on immune status.

Response to a): *Indeed, we did not associate mutational signatures with the known breast cancer subtypes but we agree this is a valuable addition. Below we have provided the requested analysis. We have excluded "normal" subtype from the statistical test since only 5 patients showed this subtype.*

AIMS subtype	n	median nr of substitutions												
		Sig 1	Sig 2	Sig 3	Sig 5	Sig 6	Sig 8	Sig 13	Sig 17	Sig 18	Sig 20	Sig 26	Sig 30	
Basal	64	680.5	33.5	3514.5	1305.5	0.0	929.5	607.5	0.0	0.0	0.0	0.0	0.0	
Her2	10	626.0	236.5	0.0	1828.5	0.0	554.5	325.0	0.0	0.0	0.0	0.0	0.0	
LumA	76	547.5	139.0	0.0	904.5	0.0	180.5	40.0	0.0	0.0	0.0	0.0	0.0	
LumB	111	676.0	158.0	0.0	1057.0	0.0	329.0	91.0	0.0	0.0	0.0	0.0	0.0	
Kruskal-Wallis		0.0259	<0.0001	<0.0001	<0.0001	0.7170	<0.0001	<0.0001	0.0172	0.1406	1.0000	0.4380	0.3797	
Post-hoc analysis*		Contrast	p	p	p	p	NA	p	p	p	NA	NA	NA	NA
		Basal v Her2	1.0000	0.0033	<0.0001	0.2553		1.0000	1.0000	1.0000				
		Basal v LumA	0.1204	0.0036	<0.0001	0.0003		<0.0001	<0.0001	0.0784				
		Basal v LumB	1.0000	<0.0001	<0.0001	0.0140		0.0577	<0.0001	0.3960				
		Her2 v LumA	1.0000	0.4510	1.0000	0.0003		0.0773	0.0034	0.0991				
		Her2 v LumB	1.0000	0.8925	1.0000	0.0026		1.0000	0.0817	0.2471				
		LumA v LumB	0.0228	1.0000	1.0000	0.8917		0.0083	0.1104	1.0000				

* p-values corrected for multiple testing (Bonferroni)

NA: Not allowed

As expected, we found signature 3 significantly enriched in the basal subtype (this signature was described as associated with ER-negatives). For the APOBEC signatures 2 and 13, the former showed a significant lower number in the basal subtype compared with others, whereas signature 13 showed higher numbers in basal and Her-2 subtypes. Finally signature 5 and to a lesser extent signature 8 tend to be lower in luminal A breast cancer. We have incorporated the signature vs subtype analysis in the results section and in a new supplemental Figure S6.

Response to b): *With regard to the request to connect recurrent driver events to immune infiltrate, this is also something we had not done as we considered it beyond the scope. Here, we analysed whether recurrently mutated genes (at least 10 observations, including substitution, rearrangement or copy-number variation data) associated with infiltrate status. The following genes were included: ARID1A, CCND1, CDH1, GATA3, MAP2K4, MAP3K1, MLL3, MYC, PIK3CA, PTEN, TP53 and ZNF217 (n=10, 32, 17, 22, 17, 20, 11, 21, 64, 13, 33 and 11, respectively), which were associated with the infiltrate status using Fisher's exact test. Even without correcting for multiple testing, p-values were all above 0.05 (respectively p=0.87, p=0.083, p=0.11, p=0.57, p=0.46, p=0.22, p=0.29, p=0.10, p=0.47, p=0.27, p=0.06 and p=0.25). One should be aware that the driver mutations in these recurrently mutated genes are mostly not one and the same mutation nor are they always categorised among the same mutational pattern complicating the interpretation of these negative results. Thus, none of the mutated genes was associated with the infiltrate status, and we have now included this in the results section in the manuscript.*

It seems to me that Figure 5 is in some sense largely independent of any analysis of mutation signatures. It shows the survival based on expression measurements, with the TIL expression determined from the lymphocyte infiltrate mixed with the tumors. Moreover, parts A and B are of doubtful utility since only one or two patients are being followed in the key groups 1 and 3. A single event could have totally changed the picture. Looking at the data of Figure 4 it appears that only a small proportion of the high TIL tumors have a large presence of the signatures 3 and 13. Thus while these signatures may be significantly correlated with high TILs, the presence of signature of the signature is unlikely to provide

much information about outcome. It might be useful for the authors to present that analysis, assuming that they have the follow up that will permit it.

Response: *Figure 5A and 5B are the evaluable cases in our own cohort, and we agree with the reviewer these are indeed low on number of events. In our earlier response to the reviewer, we stated that we included a statement in the results about the low numbers of patients at risk in our cohort. That is exactly the reason why we previously added figure 5C, where we confirm the validity in an independent, multi-centre, publically available dataset. Herein, 625 patients were included, of whom 179 developed a distant metastasis, permitting a statistically meaningful analysis.*

Upon request, we looked at the number of signature 3 and 13 substitutions in relation to outcome (RFS and OS), although this analysis is susceptible to the same argument of having low numbers of cases at risk in our cohort the reviewer rightfully touched upon. Using Cox proportional hazard's model no significant relation with outcome was found for the signatures. Thus, the reviewer is correct in assuming the outcome of patients is more complicated than just the abundance of a mutational signature.

Especially if one realises that signature 3 and 13 also contribute to the total number of mutations, of which we and others have shown that mutational load is related to poor outcome. We expect that only upon inclusion of many more cases in future studies such studies may have sufficient power to study the number of signature mutations vs outcome in depth. In the current manuscript, we were very careful in the discussion in not overestimating or overemphasizing the outcome results; just the total burden is discussed (second paragraph). Due to the limited numbers at risk in our cohort, we are reluctant to add the signature 3 and 13 vs outcome in the discussion.

We thank the reviewer for these additional views; we agree with the reviewer that there are a plethora of questions one can address in this extensive dataset. We have performed the requested analyses and added the relevant results in our main text in the results section and included an additional supplemental figure (Suppl. Fig 6).

Reviewer #4 (Remarks to the Author):

Using bioinformatics the study compares WGS data (from Nik-Zainal et al Nature 2016) and RNA-seq data for 266 breast cancer cases.

It is schematically organized in three parts.

The first part examines the expression of mutated genes to help define "driverness". Nothing much is novel but, being based on state-of-the-art methods, it should be at least comprehensive.

1 - Because emphasis seems to be made on luminal cases I am surprised that nothing is said about 8p12 amplified genes, such as FGFR1, WHSC1L1 and ZNF703.

Response: *We thank the reviewer for these valuable suggestions. With regards to point 1, 8p12 was indeed identified as recurrent amplified region, but we chose not to report on the association of the expression of genes in that locus with the amplification status, since that paragraph in the results section is centered on the driver genes identified in these samples, as reported by our collaborators (Nik-Zainal et al, Nature, 2016). In this driver list, 8p12 was the only locus (i.e. all other reported drivers were genes) and in this locus, we were unable to distinguish which of the gene(s) is or are the driver(s) and which are passengers (if any). Below, for the reviewer, we provide the expression vs amplification of the requested genes. All genes are higher expressed in the amplified cases, with strongest statistical support (Mann-*

Whitney test) for *WHSC1L1*. To meet the reviewer's request, we have now included these observations in the results section.

2 - The case of *GATA3* is interesting. Functional validation of mutated *GATA3*, such as overexpression, ChIP-seq..., would have been informative. Supervised analyses of mutated vs non-mutated cases could have been done.

Response: Our effort in this part of the text was in validating our cohort with previously reported knowledge. We did report the serendipitous *GATA3* finding, but since our study focused predominantly on the mutational signatures vs transcriptome, we regard further in depth analysis of *GATA3* beyond the scope the current study. We provided supervised analysis in the form of non-parametric tests between wild-type and mutated *GATA3* cases in the main text and in figures 2B and supplementary figure 5B. For the reviewer, we extended the analysis to test differences in transcriptomic pathways in *GATA3* mutated versus *GATA3* wild-type cases, using KEGG and BIOCARTA as pathway databases. Very few pathways were significant (multiple testing corrected and permutation p-value <0.05), and upon scrutiny of the genes in these pathways, those pathways were associated with *GATA3* being wild-type. Thus, no pathway was found significantly associated with *GATA3* mutated tumours. See below for the p-values and a representative example of genes in a pathway. Because of the lack of very significant p-values of pathways associated with *GATA3* mutation, we decided to not include these results in the current manuscript.

Database	Pathway	permutation pvalue	Bonf-Holm pvalue
Biocarta	CCR3 signaling in Eosinophils	0.017	0.016
Biocarta	Telomeres Telomerase Cellular Aging and Immortality	0.020	0.016
Biocarta	Ion Channels and Their Functional Role in Vascular Endothelium	0.042	0.016
Biocarta	TPO Signaling Pathway	0.048	0.016
Biocarta	CXCR4 Signaling Pathway	0.048	0.019
Biocarta	Apoptotic Signaling in Response to DNA Damage	0.048	0.019
KEGG	Propanoate metabolism	0.008	0.007
KEGG	Taste transduction	0.028	0.008
KEGG	Glycosphingolipid biosynthesis neolactoseries	0.028	0.008
KEGG	Caprolactam degradation	0.033	0.008
KEGG	Regulation of actin cytoskeleton	0.034	0.008
KEGG	Glycine serine and threonine metabolism	0.040	0.008

3 - Please include a table with P53 mutations separating the two potential types, expressed/oncogenic vs absent/TSG. Relate this to 17p LOH.

Response: Supplementary figure 3 contains TP53 expression versus type of mutations, and as requested we checked the LOH and copy-number neutral (cnn) LOH status, and now included the latter in the results section and to supplementary figure 3 since this was the most informative. Below are both tables. The group of mutant cases showing significantly higher TP53 expression is the “missense” group, and of the 18 cases without LOH, 11 are of the missense type. However, analysis of cnnLOH status was informative (Fisher’s exact $p=0.021$); evaluating the expected versus observed cases with cnnLOH, the missense cases showed half the number of cases with cnnLOH (yellow) than expected while the cases with nonsense mutations had almost double the number of cases with cnnLOH (green) than expected. Of the in total 100 missense cases, 91% did not have cnnLOH and this all is in line with the theory that a dominant mutation does not require an additional event.

Effect	TP53_LOH		Effect	cnnLOH_TP53		Total
	no	yes		no	yes	
Deletion	0	4	Deletion	3	1	4
Tandem duplication	1	2	Tandem duplication	2	1	3
Translocation	0	2	Translocation	1	1	2
complex sub	0	3	complex sub	2	1	3
ess splice	1	17	ess splice	15	3	18
frameshift	0	33	frameshift	25	8	33
HD	0	1	HD	1	0	1
inframe	0	6	inframe	5	1	6
missense	11	89	missense	91	9	100
nonsense	5	27	nonsense	21	11	32
WT	165	133	Total	166	36	202

Numbers in parenthesis are expected numbers.

The second part is about proliferation.

4 - It would be interesting to relate proliferation/mitotic cycle with mutations/amplifications/deletions, and to signatures. For examples in Fig 3, is there a gradient of MYC, CCND1... amplifications (more at right)? It would be interesting to add ER- cases.

Response: To study the relation of driver genes mentioned in the main text to cell cycle, we investigated *AKT1*, *CCND1*, *CDH1*, *ERBB2*, *GATA3*, *IGF1R*, *MAP2K4*, *MDM2*, *MYC*, *PIK3CA*, *PTEN*, *TP53* and *ZNF217* (substitution, rearrangement or copy-number variation data were included and Fisher's Exact test was used). Of the drivers studied, *CCND1*, *MYC* and *ZNF217* amplification and *TP53* mutation ($p < 0.0001$, $p < 0.0001$, $p = 0.002$ and $p < 0.0001$, respectively) were significantly enriched in the top quartile with highest expression of cell-cycle regulated genes.

Although significant, these are all well known observations and we believe we are not increasing knowledge by stating that e.g. cyclin D1 is associated with cell-cycle related pathways. Thus in our view, these results do not add much to the observation of cell-cycle related gene expression versus the mutational burden of a cancer, which was the reason for including figure 3. However, to meet the reviewer on this point, we have added text in the second paragraph of the discussion section on this subject.

Regarding the ER-negative cases: figure 3 is a direct consequence of the pathway results, where cell-cycle was found associated with mutational burden in ER-positive breast cases only. So at this point there

is no precedent to reintroduce the ER-negatives here. Additional arguments for the exclusion of ER-negatives are presented at point 6.

The third part is about mutational burden, mutational signatures and immune response. In the absence of further validation studies on mutated/amplified/deleted genes, and given the recent availability of immune-based treatments, this part is actually the most (only?) interesting.

5 - What is the importance of mutational burden rather than mutational type is not clear in the written text. This point should be made clearer.

Response: *We agree with the reviewer that this part is indeed the most important one. Regarding point 5, the mutational burden in a sample is the sum of mutations, but consisting of a mix of mutational types. By untangling the type of mutation, we were able to investigate the signatures to the transcriptome. The total burden in itself was also investigated. To meet reviewer's comment, we clarified the importance of mutational burden in the discussion, where in the complete second paragraph the relevance to cell-cycle and outcome is detailed and put into context. Furthermore, we now also elaborate in the concluding paragraph in the discussion on the role of mutational burden in relation to changes in the transcriptome and immune response.*

6 - The analysis should not have been limited to ER+ cases. This is really surprising because immune signatures have been shown to be important in basal/triple negative/medullary cases (e.g. Teschendorff et al., LCK metagene...). There is much to do here. E.g. samples classified upon these expression signatures may be examined for mutational signatures.

Response: *It is true that ER-negatives have been largely excluded from the results; this is first and foremost due to the fact that we were following up on the significant results we identified in the pathway analysis. Additionally, we would like to mention that, as indeed known from literature and also observed in our dataset, virtually all ER-negative patients show high levels of infiltrating lymphocytes. In our cohort among the ER-negative cases just 4 of them have the status of "nill infiltrate", making it impossible to do meaningful analyses, since we lack a contrast group with sufficient numbers to assure statistical valid analyses. Comparing the ER-negative high infiltrate group with ER-positives with low infiltrate will be highly confounded by the all-pervasive transcriptomic differences between ER-negative and positive breast cancers. Finally, similar arguments in terms of mutational burden and cell cycle/proliferation are applicable for the ER-negatives; the vast majority of ER-negative cases has a high mutational load and are actively cycling; the active cell-cycle has been known from literature for many years (amongst others, Perou et al, Nature, 2000; Van't Veer et al, Nature, 2002; Wang et al, Lancet, 2005). Regarding mutational load, just 2 ER-negative cases are among the bottom 20% after ranking all cases on mutational load (stated in methods), and, not unexpectedly, 88% of ER-negatives are grade III, with no grade I among the ER negatives in our cohort. So, although it is obvious these processes are tightly associated with ER-negativity, it also prohibits identifying expression differences due to the lack of a sizeable contrast group. We have now included these arguments in the discussion.*

7 - There is a way to establish a BRCAness score (Popova et al., 2012). Please test it and see if it matches mut S3. Mut S3 seems to be associated with ER-, which BRCA1 are. Again, test this.

Response: *Very probably unknown to the reviewer due to the very recent publication date, this point is fully covered in the first paper of our consortium (Nik-Zainal et al, Nature 2016, Figure 5, page 52). It describes in detail the BRCAness of the samples we also used for our transcriptome analyses. In the Nik-Zainal paper the integrated analysis of copy-number, substitution and rearrangement signatures showed 7 consensus clusters, which was also able to distinguish BRCA1(ness) as a separate group from BRCA2(ness), see figure 5, clusters D and G of the Nik-Zainal paper. That figure also shows the indeed clear association between ER-negativity, substitution signature 3 and rearrangement signature 3 with BRCA1 (and BRCA1- like samples; e.g. BRCA1 methylated BC cases); and ER-positivity and rearrangement signature 5 with BRCA2 (and BRCA2-like samples). Since these results are published, there is no rationale to restate them in our manuscript.*

8 - Concerning prognosis, were multivariate analyses done?

In conclusion, the study is timely and potentially interesting but as such, is a bit superficial.

Response: *We agree with reviewer that multivariate analysis is of added value but the numbers of patients at risk in the current cohort are too low to permit a multivariate analysis. However, we did analyse in the validation cohort if the TIL signature was independent of the cell cycle signature associated with outcome. This multivariate analysis clearly shows TIL is, independently from proliferation, associated with prognosis. We have included this multivariate data to the results section.*

Cox HR (MFS)	Univariate		Multivariate	
	HR (95% CI)	p-value	HR (95% CI)	p-value
TIL signature	0.527 (0.36-0.78)	0.001	0.534 (0.36-0.79)	0.002
MCC signature	1.832 (1.34-2.50)	<0.0001	1.811 (1.33-2.47)	<0.0001

In summary, we thank the reviewer for the suggestions and comments and we agree that with the current rich multidimensional datasets, indeed many questions can be asked. For the current manuscript, we focused on analysing the global transcriptome in relation to the known substitution signatures since this is the first dataset with sufficient numbers of signature mutations per breast cancer in which such a question can be addressed. However, various points raised by the reviewer pertained to the part where we described and validated our cohort, and although we concur these points are indeed of interest, they are of little added value to the main message of our manuscript. We performed all requested analyses (indirectly in case of point 7) where possible, and added these to the results and discussion sections and supplemental figures 3 and 5B.

To meet the editors' specific request we here list the changes we made to the manuscript following the advice and comments of reviewer #1:

Introduction:

-We expanded the introduction to provide context for the mutational signatures. We included references to other studies and to our consortium's 560 whole-genome paper which is now published.

Results:

-We rewrote, extended and clarified a paragraph in the results section to verify our cohort is comparable with previously described breast cancer cohorts.

-We included additional references in the results section and clarified methods for figure 1 and the pathway analyses.

- We added the following analyses reviewer #1 requested to the results section:

- analysed the signatures proportionally

- evaluated the prognostic effect of the TIL signature in an independent cohort

- performed additional tests for trend with respect to signature 13/2 and infiltrate

- updated Figure 1 (new cluster-group)

- added results related to the observations for the CCND3 and GATA3 genes as a new Figure 2

Discussion:

-We rewrote the discussion to elaborate on various aspects; the second paragraph on mutational burden was added, which also contains context to both replication dependent/independent signatures contributing to aggressiveness.

-We extended the discussion on the relation of signature 3 and BRCA1/2-ness.

-We rewrote the part on the association of the signatures 3 and 13 and immune response and clarified the relevance.

Finally, to accommodate the additional analyses the reviewers requested, we have added a panel to supplementary Fig. 3, replaced former supplementary Fig.6 with a new one (molecular subtypes) and combined former Supplementary Figures 8,9 and 10 to 3 panels in Supplementary Fig. 8.

Thus, in order to incorporate the comments and suggestions of all reviewers, we have extensively rewritten the manuscript.

Reviewers' Comments:

Reviewer #3 (Remarks to the Author):

The multiple rounds of review for this paper have improved the description of what was actually done and what can be claimed. For the purposes of this review I am going to accept that all of the measurements and analyses are correct and try to address the overriding question of what contribution this paper makes to the understanding of cancer.

As the other reviewers have said this paper has several parts, but only one is mentioned in the title-the "impact" of specific mutational signatures on the immune response in luminal breast cancer. The highlight of the abstract is the statement that the data "suggest" the impact is dependent on specific mutational signatures, although squishier words are used. This paper will likely be cited prominently based on the title and abstract, which will generate an ongoing impression in the literature that is somewhat off the mark. The actual data in Figure 4, shows that while mutational signatures may be correlated with immune response, the correlation is not strong enough to reverse the inference and use the presence of a signature to determine if a tumor had high or low immune response. One might want the title of the paper to be something like "Weak association of immune response with mutational signatures in luminal breast cancer". But really one might want a title that deals with the entire scope of the paper.

Continuing with the immunology, I am uncertain with the presentation of the logic. I understand the analysis of the charge changes for the different substitutions, but I found no information in the paper about how these were quantitatively manipulated so that charge effects were assigned to mutational signatures. But more importantly I do not see why signatures were used at all. Why not just look at the charge changes on the individual mutations that occurred in each tumor and classify the tumors in order of their increased number of positive changes (or whatever measure seems reasonable)? If charge really is important, should that analysis not that give a cleaner view of it? The fact that the investigators may have come across the possibility by thinking of mutational signatures does not mean they should remain wedded to the signature concept. Why not focus more cleanly on what might be an important issue? But the fact that charge on the AA's is critical is just one possibility. The mutations could be doing something else entirely, for example affecting members of a large group of specific genes, each at a frequency too low to attract attention as a driver gene.

The outcome data in Figure 5 seems to me to exist in a world where mutational signatures play no role. As presented they are just expression measurements-- the TIL signature could be replaced by histological examination of the tumors to judge TIL activity, and cell cycle by other measures of proliferation. There are many publications in the literature about this and I assume some are much like Figure 5c.

So after reading the paper now several times in different forms, my take home message is that the concept of mutational signatures does not give much leverage in studies of the transcriptome. I think the concept of mutational signatures is strong and relevant when dealing with specific mutational mechanisms and repair defects and their direct consequences in DNA. But there are large increases in biological complexity as one moves from DNA sequence to expression and finally to biological outcome. And it seems to me the data demonstrate that the signatures do not provide much in the way of an organizing principle to deal with these consequences of mutation. I note that there has been a recent paper in PLOS concerning the over-interpretation of epigenetic studies (citation below) leading to excitement that may now be decaying.

In my view of science a statement of the limitations of the signature concept would be a perfectly reasonable message for a paper, a message that does not have to be hidden by an effort to find something new in biology. Moreover if the signature concept allows efficient re-discovery of things that are already known after past laborious efforts, then that would also be an interesting message since it may have considerable importance when applied in a new area.

Reference: Birney E, Smith GD, Grealis JM (2016) Epigenome-wide Association Studies and the Interpretation of Disease -Omics. PLoS Genet 12(6): e1006105.
doi:10.1371/journal.pgen.1006105

Reviewer #4 (Remarks to the Author):

The authors have correctly answered questions and suggestions.

Reviewers' comments:

Reviewer #3 (Remarks to the Author):

The multiple rounds of review for this paper have improved the description of what was actually done and what can be claimed. For the purposes of this review I am going to accept that all of the measurements and analyses are correct and try to address the overriding question of what contribution this paper makes to the understanding of cancer.

As the other reviewers have said this paper has several parts, but only one is mentioned in the title-the "impact" of specific mutational signatures on the immune response in luminal breast cancer. The highlight of the abstract is the statement that the data "suggest" the impact is dependent on specific mutational signatures, although squishier words are used. This paper will likely be cited prominently based on the title and abstract, which will generate an ongoing impression in the literature that is somewhat off the mark. The actual data in Figure 4, shows that while mutational signatures may be correlated with immune response, the correlation is not strong enough to reverse the inference and use the presence of a signature to determine if a tumor had high or low immune response. One might want the title of the paper to be something like "Weak association of immune response with mutational signatures in luminal breast cancer". But really one might want a title that deals with the entire scope of the paper.

Response:

It is correct that our paper has several parts and only one is mentioned in the title. However, with only 15 words available for the title it is difficult to capture the paper's content entirely. Therefore, we chose to mention the main novelty. Of note, we provide results of the relation between immune infiltrate and signatures by using expression data (several Biocarta pathways and an independent TIL signature) but also using pathological infiltrate status, which we think justifies our use of the current title. However, the concerns of the reviewer of a too optimistic title are noted. We have changed the title to:

Integration of the breast cancer genome and transcriptome associates specific mutational signatures with immune response

Continuing with the immunology, I am uncertain with the presentation of the logic. I understand the analysis of the charge changes for the different substitutions, but I found no information in the paper about how these were quantitatively manipulated so that charge effects were assigned to mutational signatures.

Response:

The reviewer indicates that we did not properly describe how the charge changes were quantified. In fact; we have stated in the Methods: "Changes in electric charge were evaluated by assigning Aspartic acid and Glutamic acid as negatively charged and Lysine, Arginine and Histidine as positive, followed by looking at the amino-acid change caused by the substitution". To meet reviewer's comment we have extended this into: "Changes in electric charge were evaluated by assigning Aspartic acid and Glutamic acid as negatively charged and Lysine, Arginine and Histidine as positive. Next, base substitutions were evaluated if a resulting amino-acid change led to a change in electrical charge. Since all substitutions are assigned to a mutational signature, the total number of charge changing (and hydrophobicity changing) substitutions per signature was established."

But more importantly I do not see why signatures were used at all. Why not just look at the charge changes on the individual mutations that occurred in each tumor and classify the tumors in order of their increased number of positive changes (or whatever measure seems reasonable)? If charge really is important, should that analysis not that give a cleaner view of it? The fact that the investigators may have come across the possibility by thinking of mutational signatures does not mean they should remain wedded to the signature concept. Why not focus more cleanly on what might be an important issue? But the fact that charge on the AA's is critical is just one possibility. The mutations could be doing something else entirely, for example affecting members of a large group of specific genes, each at a frequency too low to attract attention as a driver gene.

Response:

To recapitulate our rationale: pathway analysis showed that immune related expression patterns associated with certain signatures. We then explored properties of these signatures to investigate why the association with immune response could be different between signatures, and electrical charge was one of these properties.

The reviewer suggests to investigate total change in charge, irrespective of the signatures. We agree with the reviewer that this analysis makes sense although it does not answer our question why only some signatures associate with immune response. However, we have now done the requested analysis which shows that patients with an increasing lymphocytic infiltrate show higher numbers of substitutions leading to an increase in electric charge ($p < 0.0001$, see boxplot below). Even after excluding APOBEC substitutions from this analysis, the significance between increased charge and lymphocytic infiltrate remained. This further hints that mutations leading to an increase in charge increase the potential that the tumor-cell is sensed as non-self. And, as explained in the discussion, signature 2 and 13 are very efficient in generating mutated peptides with increased charge due to the sort of substitutions they generate. We incorporated the abovementioned analysis in the results and discussion.

*Lastly, nowhere in the paper do we declare that the charge is 'critical' or the definitive answer to the relation of signature 13 mutations and immune response. We have explained in the discussion why some signatures have a high percentage of charge changing amino-acid substitutions, and we kept this part. However, to meet the reviewer's concerns, we have removed the underlined part in the sentence in the discussion and added 'may' (bold): "...we observe a signature specific association with immune response, possibly arising from an increased number of more positively charged neo-antigens that consequently **may** stimulate the immune response more effectively".*

In addition, a statement was added to acknowledge the reviewer's remark that confirmation of our findings as well as studies providing mechanistic support are necessary.

group	n	Mean	Median
nil	43	7.7	4.0
mild	211	9.9	6.0
mod_severe	94	17.1	12.0

The outcome data in Figure 5 seems to me to exist in a world where mutational signatures play no role. As presented they are just expression measurements-- the TIL signature could be replaced by histological examination of the tumors to judge TIL activity, and cell cycle by other measures of proliferation. There are many publications in the literature about this and I assume some are much like Figure 5c.

Response:

We agree the TIL signature is used as a surrogate but the current dataset with 560 complete genomes and 266 transcriptomes does not have well-annotated outcome data. We have stated in our previous response to this reviewer on this issue that there are too few cases with known mutational signatures and clinical follow-up to provide a well-founded answer of the prognostic role of mutational signatures and that for this analysis more cases are needed. Therefore, instead we infer using the TIL signature as a surrogate for the mutational signature, its prognostic value in a well-annotated publically available dataset. We also feel that providing these results is completely in line with the scope of the paper and for the common expectation to present the potential clinical significance of the current finding.

So after reading the paper now several times in different forms, my take home message is that the concept of mutational signatures does not give much leverage in studies of the transcriptome. I think the concept of mutational signatures is strong and relevant when dealing with specific mutational mechanisms and repair defects and their direct consequences in DNA. But there are large increases in biological complexity as one moves from DNA sequence to expression and finally to biological outcome. And it seems to me the data demonstrate that the signatures do not provide much in the way of an organizing principle to deal with these consequences of mutation. I note that there has been a recent paper in PLOS concerning the over-interpretation of epigenetic studies (citation below) leading to excitement that may now be decaying.

In my view of science a statement of the limitations of the signature concept would be a perfectly reasonable message for a paper, a message that does not have to be hidden by an effort to find something new in biology. Moreover if the signature concept allows efficient re-discovery of things that are already known after past laborious efforts, then that would also be an interesting message since it may have considerable importance when applied in a new area.

Reference: Birney E, Smith GD, Grealley JM (2016) Epigenome-wide Association Studies and the Interpretation of Disease -Omics. PLoS Genet 12(6): e1006105. doi:10.1371/journal.pgen.1006105

Response:

The reviewer indicates the effect of the various signatures on the transcriptome is limited. We indeed agree that the biological complexity is high in these analyses and acknowledge the cited paper, but we deem over-interpretation not a major issue, since we evaluated few signatures with pathways only and this analysis was corrected for multiple testing. So we did not test thousands of independent genes with a multitude of endpoints.

We do show in our paper that transcriptomic expression patterns associate with certain mutational signatures. We show this e.g. in our pathway analysis, but also – upon suggestion of the reviewer - the molecular subtypes as defined by expression data show clear associations with the type of mutational signature. Also, previously others have reported on the link of mutational burden and proliferation, thus showing that there are validated effects of mutational signatures on the transcriptome. Thus, we provide evidence that the mutation signatures can have impact on the transcriptome. Since we are the first, to our knowledge, to associate global gene expression data to whole-genome mutational signatures in this manner, we recognise the need for independent validation.

Lastly, we do realise that the 12 currently identified signatures are only one possible way of categorising changes in DNA, but when using the current signatures, as we did, the observations reported in this paper can be made.